# Spatial variability in Alpine reservoir regulation: deriving reservoir operations from streamflow using GAMs

Manuela Irene Brunner[1,2,3] and Philippe Naveau[4]

[1]WSL Institute for Snow and Avalanche Research SLF, Davos Dorf, Switzerland
[2]Institute for Atmospheric and Climate Science, ETH Zurich, Zurich, Switzerland
[3]Climate Change, Extremes and Natural Hazards in Alpine Regions Research Center CERC, Davos Dorf, Switzerland
[4]Laboratoire des Sciences du Climat et l'Environnement (LSCE, EstimR) CNRS, Gif-sur-Yvette, France

**Correspondence:** Manuela I. Brunner (manuela.brunner@env.ethz.ch)

**Abstract.** Reservoir regulation affects various streamflow characteristics from low to high flows, with important implications for downstream water users. However, information on past reservoir operations is rarely publicly available and it is hardly known how reservoir operation signals, i.e. information on when water is stored in and released from reservoirs, vary over a certain region. Here, we propose a statistical model to reconstruct reservoir operation signals in catchments without information on reservoir operation. The model uses streamflow time series observed downstream of a reservoir that encompass a period before and a period after a known year of reservoir construction. In a first step, a generalized additive model (GAM) regresses streamflow time series from the unregulated pre-reservoir period on four covariates including temperature, precipitation, day of the year, and glacier mass balance changes. In a second step, this GAM, which represents natural conditions, is applied to predict natural streamflow, i.e. streamflow that would be expected in the absence of the reservoir, for the regulated period. The difference between the observed regulated streamflow signal and the predicted natural baseline should correspond to the reservoir operation signal. We apply this approach to reconstruct the seasonality of reservoir regulation, i.e. information on when water is stored in and released from a reservoir, from a dataset of 74 catchments in the Central Alps with a known reservoir construction date (i.e. date when reservoir went into operation). We group these reconstructed regulation seasonalities using functional clustering to identify groups of catchments with similar reservoir operation strategies. We show how reservoir management varies by catchment elevation and that seasonal redistribution from summer to winter is strongest in high-elevation catchments. These elevational differences suggests a clear relationship between reservoir operation and climate and catchment characteristics, which has practical implications. First, these elevational differences in reservoir regulation can and should be considered in hydrological model calibration. Furthermore, the reconstructed reservoir operation signals can be used to study the joint impact of climate change and reservoir operation on different streamflow signatures, including extreme events.

## 1 Introduction

Reservoir regulation affects various streamflow characteristics – including variability (Eisele et al., 2004; Ferrazzi et al., 2019), seasonality (Biemans et al., 2011; Adam et al., 2007; Rottler et al., 2019), and extreme events (Verbunt et al., 2005; He et al., 2017; Wang et al., 2017; Wan et al., 2017; Vicente-Serrano et al., 2017; Mahe et al., 2013; Tijdeman et al., 2018; van Oel

et al., 2018; Volpi et al., 2018; Brunner, 2021) – in almost 50% of the world's large rivers (>1000 m$^3$ s$^{-1}$) and in 8% of the rivers overall (Lehner et al., 2011). Regulation patterns may vary across regions and hydro-climates as reservoirs are operated for different purposes including irrigation, energy production, water supply, and recreation, in some cases in a multi-purpose way (Lehner et al., 2011; Brunner et al., 2019a). However, information on these reservoir operation signals, i.e. on when water is stored in and when it is released from reservoirs, is hardly publicly available, despite its importance for model calibration and impact assessments (Yassin et al., 2019; Speckhann et al., 2021; Brunner et al., 2021; Steyaert et al., 2022). In some cases, reservoir operation records are made available by the operating agencies (e.g. Steyaert et al., 2022), however, this is the exception rather than the rule. As a consequence, it is often unclear how reservoir regulations vary across a region and whether and how the regulation patterns are related to catchment characteristics – knowledge that might be useful to transfer reservoir regulation information to basins without such information. Because of the lack of reservoir regulation information, hydrological and land-surface models often use generic reservoir operation schemes that do not necessarily reflect local behavior, which is particularly problematic when simulating streamflow at sub-monthly resolution or when modelling extreme events (Hanasaki et al., 2006; Yassin et al., 2019; Turner et al., 2021).

Various attempts have been made to infer reservoir operation signals from different types of data sources including optical and altimetry remote sensing (Peng et al., 2006; Eldardiry and Hossain, 2019; Hou et al., 2022; Du et al., 2022), reservoir purpose, simulated inflows and water withdrawals (Hanasaki et al., 2006; Voisin et al., 2013) or in- and outflows (Turner et al., 2021). To identify the time scales most affected by reservoir operation, White et al. (2005) and Shiau and Huang (2014) used the wavelet transform on both observed in- and outflow time series and compared their wavelet power spectra. To estimate reservoir release policies, Coerver et al. (2018) used fuzzy rules to link inflow and storage with reservoir release for a set of reservoirs in Asia and North America and Turner et al. (2021) developed harmonic regression models using observed and simulated daily reservoir in- and outflows for large reservoirs in the continental United States (Steyaert et al., 2022). To map input–output relationships for dams around the world, Ehsani et al. (2016) used artificial neural networks and data on inflow, release and storage. While these approaches are very helpful for reservoir signal reconstruction in case both in- and outflow data are available, inferring the reservoir operation signal based on outflow information only remains challenging.

Here, we propose a statistical three-step approach for reservoir signal reconstruction in catchments where reservoir outflow time series are available. The approach is based on a generalized additive model (GAM) that enables reconstructing reservoir operation signals from observed streamflow time series measured downstream of a reservoir or a set of reservoirs and encompassing the period before and after a known year of reservoir construction. In a first step, the approach fits a GAM to streamflow observations representing natural pre-reservoir conditions using precipitation, temperature, day of the year, and glacier mass balance changes as covariates. In a second step, this GAM is applied to covariates derived for the regulated post-dam period to predict natural streamflow for this regulated period. In a last step, the reservoir regulation signal is reconstructed by subtracting the predicted 'natural signal' from the observed regulated signal. This resulting signal indicates how much water is stored in and released from reservoirs in which season (i.e. day of the year). These reservoir-storage-seasonality signals take a reservoir perspective and provide information on storage in addition to releases, but not on inflow. Therefore, they are distinct from the signals extracted through other approaches, e.g. simulated water releases (Coerver et al., 2018); spectral differences between

in- and outflows highlighting the time scales most affected by reservoir regulation (White et al., 2005; Shiau and Huang, 2014);
or water storage and release policies, which define release decisions as percent deviations from long term mean inflow (Turner et al., 2021). Our approach can be used to reconstruct reservoir operation signals in catchments where streamflow and climate data are available for a period before and after a known date of reservoir construction. Such information is more widely available than reservoir in- and outflow measurements, which means that the approach is applicable in different regions around the globe where streamflow observations and information on reservoir construction dates are available.

Here, we use the proposed approach to shed light on spatial variations in reservoir regulation signals and their relationship to catchment characteristics. To do so, we apply the approach to extract reservoir signals from observed time series of 74 catchments in the Central Alps (Section 2.2). From this database of 74 extracted signals, we identify groups of catchments with similar reservoir operation strategies using functional data clustering (Section 2.3) (Chebana et al., 2012; Ternynck et al., 2016). The functional form is derived from discrete observations (Ramsay and Silverman, 2002) either by smoothing the data non-parametrically (Jacques and Preda, 2014) or by projecting the data onto a set of basis functions. The basis function (e.g. B-spline, Fourier, or wavelet bases) coefficients can be used for clustering (Cuevas, 2014). It has been shown in previous studies that functional data representations can be beneficial to identify groups of similar hydrographs over a range of temporal scales, such as spring flood events (duration of six months; Ternynck et al., 2016), flood events (duration of several days; Brunner et al., 2018), low flow events (Laaha et al., 2017), diurnal discharges (duration of one day; Hannah et al., 2000), yearly hydrographs (Merleau et al., 2007; Jamaludin, 2016), and streamflow regimes (Brunner et al., 2020). Here, we use functional data clustering to identify groups of catchments with similar reservoir operation seasonalities. We then assess whether and how catchments with different reservoir operation strategies differ in their location and catchment characteristics. The combination of the proposed reservoir signal reconstruction approach with functional clustering allows us to provide insights into how reservoir regulation varies spatially in the Alps and to which degree these variations are related to catchment characteristics.

## 2 Methods

### 2.1 Dataset

The Central Alps are an interesting region to study different reservoir regulation patterns because this region is characterized by diverse hydro-climatic regimes (Bard et al., 2015), which are often heavily influenced by reservoirs (Lehner et al., 2005; Brunner et al., 2019a). Therefore, we identify a large sample of 74 regulated catchments in the headwater regions of the four major European rivers originating in the Central Alps, namely, the Rhine, Rhône, Danube, and Po for which the date of reservoir construction, i.e. date when reservoir went into operation, is known and for which observed daily streamflow data are available for both a period before and a period after reservoir construction (Figure 1). The observed streamflow time series were obtained from national agencies in Switzerland (Federal Office for the Environment, FOEN), Austria (Austrian Ministry of Sustainability and Tourism), and eastern France (Banque HYDRO) and regional agencies in southern Germany (regions Bavaria [Bavarian State Office for the Environment] and Baden-Württemberg [State Institute for the Environment Baden-Württemberg]). The streamflow records of the different catchments do not necessarily cover the same time period, however,

each catchment has streamflow data for at least 10 years before and after reservoir construction (see Figure 2 for an example time series in the Swiss Alps). Northern Italy was excluded from the analysis because streamflow records provided by the regional agencies did not cover the pre-reservoir construction period.

In addition to streamflow, we derive daily meteorological time series (precipitation and temperature) for each catchment from the gridded observational E-OBS dataset at 25 km spatial resolution for the period 1950–2020 (Cornes et al., 2018) by averaging over all grid cells within a catchment. If present, missing values in the time series of all variables are replaced by the daily mean over the natural period for the natural data and over the regulated period for the regulated data. Temperature and precipitation time series are smoothed over a moving time window of 5 days to remove noise because smoothing improves

model performance. Furthermore, data on reservoir locations and construction dates are also obtained from national agencies (Switzerland: FOEN, Austria: Austrian Ministry of Sustainability and Tourism, France: Comité Francais des Barrages et Reservoirs (https://www.barrages-cfbr.eu/)) and open source databases (Germany: Speckhann et al. (2021)). To account for changes in glacier melt contributions over time, we compute annual glacier mass balance changes for each of the selected catchments using simulated mass balance changes over the period 1951–2020 for the glaciers in the Randolph Glacier Inventory (RGI

Consortium, 2017; Compagno et al., 2021). After estimating the average mass balance change for each glacier in a catchment by weighting changes across different elevation bands, each annual mass balance time series is dis-aggregated into daily resolution by smoothing the annual signals over 365 days. This smoothing avoids step-like features in the mass balance change time series.

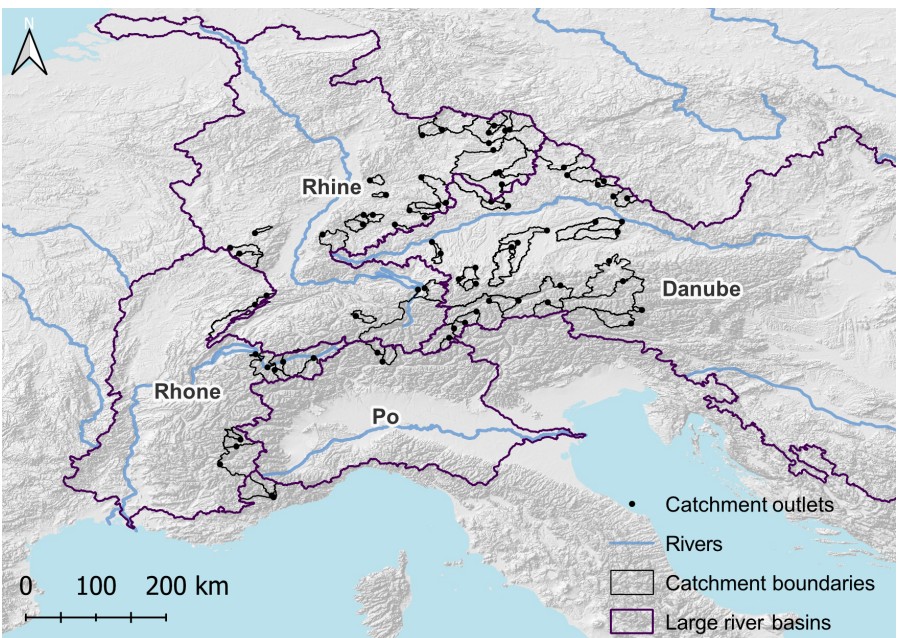

**Figure 1.** 74 catchments in the Central Alps with at least 10 years of streamflow data before and after reservoir construction (black catchment outlines).

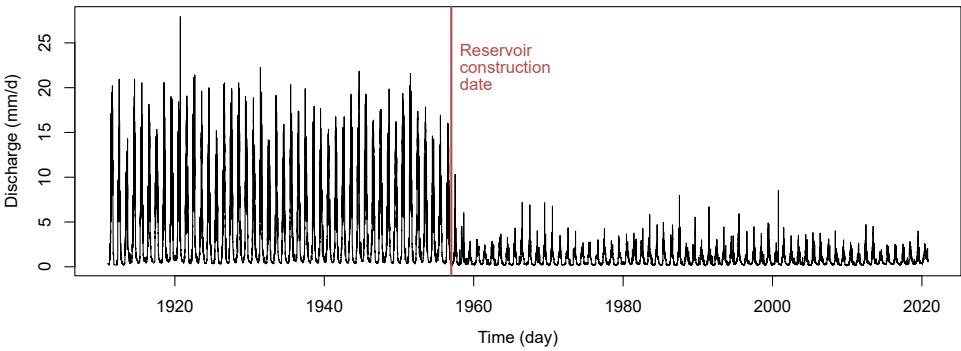

**Figure 2.** Streamflow time series for the catchment of the Drance de Bagnes (gauge Le Châble) illustrating streamflow changes induced by the construction of the Mauvoisin reservoir in 1957.

## 2.2 Reservoir signal reconstruction using GAMs

Here, we propose a modelling approach to reconstruct the reservoir operation signal from observed streamflow time series measured downstream of a reservoir before and after reservoir construction, representing natural and regulated conditions, respectively (Figure 3a,b). Before the reservoir construction date, a regression scheme can learn the natural link between streamflow times series and some appropriate meteorological explanatory variables. In this work, this natural baseline signal is obtained by applying a generalized additive model (GAM; Hastie and Tibshirani, 1986) during the pre-reservoir time period.

After the reservoir construction, the reservoir operation signal can be defined as the difference between the regulated streamflow time series and the signal that would have been measured without the reservoir. The latter signal was never observed but it can be estimated by applying the learning GAM link to post-reservoir meteorological explanatory variables.

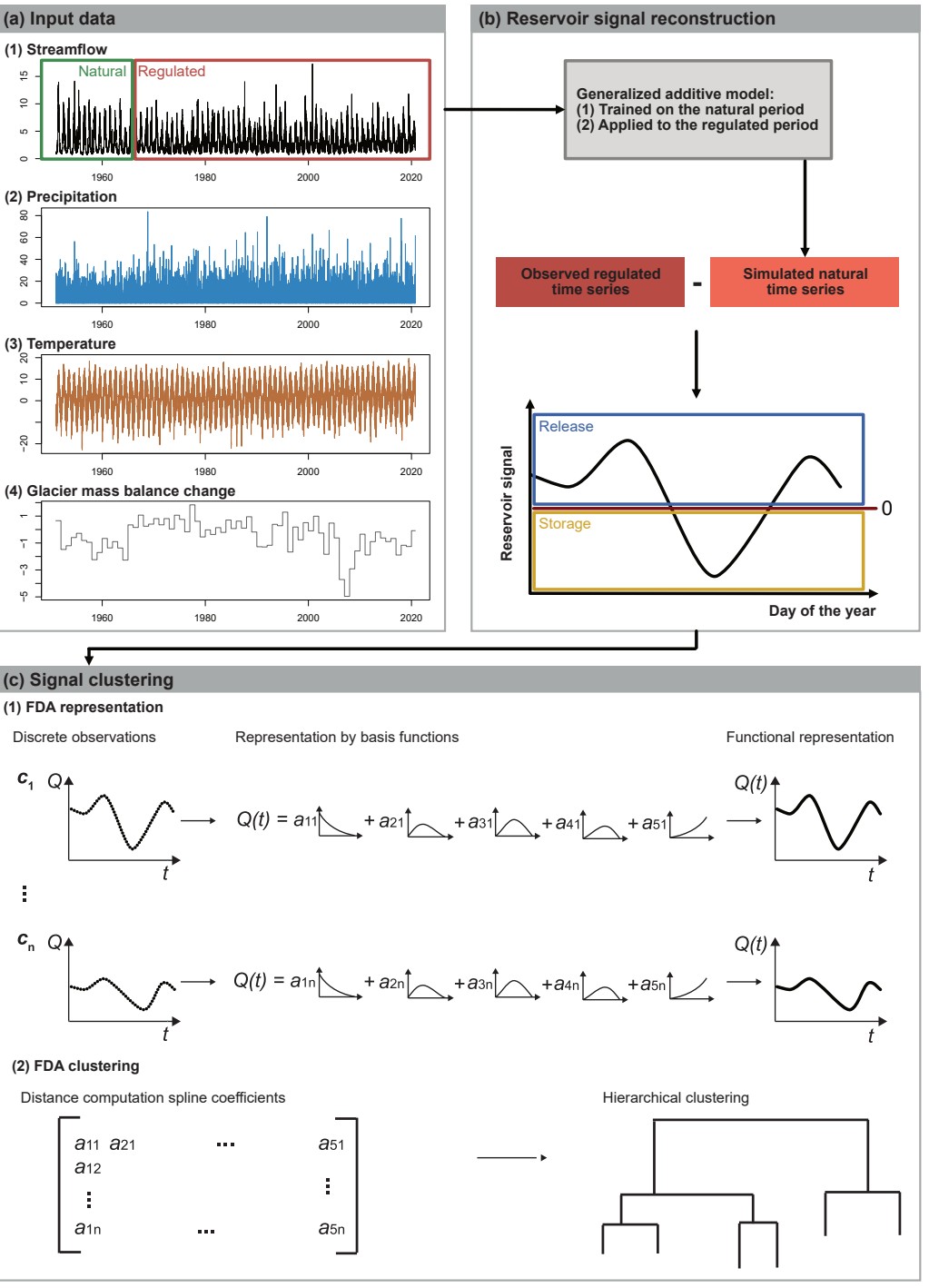

**Figure 3.** Workflow illustration: (a) Input data used to fit and run the generalized additive model (GAM): streamflow, precipitation, temperature, and glacier mass balance changes for a period before and after reservoir construction representing natural and regulated conditions, respectively; (b) GAM fitting using the natural data before reservoir construction, GAM use to predict the natural signal for the regulated period, and reconstructing the regulation signal by subtracting the predicted 'natural time series' from the observed regulated time series. That is, determining the seasons with reservoir storage and release; and (c) reservoir signal clustering using functional data analysis (FDA) using hierarchical clustering on functional representations (i.e. spline basis functions) of the reconstructed signals of all catchments in the dataset.

Generalized additive models (GAMs) extend the linear regression setup and therefore represent a flexible model structure to predict streamflow. The classical additive linear link, $\sum \beta_j X_j$, between the observational vector $Y$ and the explanatory variables $(X_1, \ldots, X_p)^T$ is replaced by a sum of smooth functions $\sum f_j(X_j)$ (see e.g. Hastie and Tibshirani, 1986). Hence, GAMs represent nonlinear relationships between covariates and the target variable. Each smooth function $f_j(.)$ corresponds to a linear projection on a given basis, here a cubic smoothing spline representation (see e.g. Hastie and Tibshirani, 1986; Wood, 2017). Typically, a GAM is written as:

$$y_t = \sum_{j=1}^{p} f_j(x_{tj}) + \sigma\, \epsilon_t, \tag{1}$$

where $\sigma > 0$ and $\epsilon_t$ represents a standardized random noise. In this study, the response variable $y_t$ corresponds to streamflow time series in mm/d (units). Alternatively, GAMs have in the context of reservoirs also been used to predict other variables than streamflow such as eutrophication levels (Catherine et al., 2010) or downstream water temperatures (Coleman et al., 2021). The index $t$ represents the time evolution in days and spans the time period before reservoir construction, which varies by catchment. For example, the construction of the Mauvoisin reservoir in 1957 can be clearly identified in the streamflow time series of the catchment Drance de Bagnes (gauge Le Châble, Figure 2). In this study, the set of explanatory variables, $(X_1, \ldots, X_p)^T$, contains three climatological parameters including temperature, precipitation, seasonality (day of year), and modelled glacier mass balance changes. During the unregulated pre-reservoir period, the GAM learns the non-linear relationship between streamflow time series and corresponding climatological parameters. Then, the estimated transfer function $f_j$ calibrated on the unregulated period is applied via equation (1) to the regulated period to predict natural streamflow. The approach relies on the assumption that the relationship between climate variables and streamflow remains stable over time. The main advantage of GAMs is that cubic spline modeling offers flexibility for each covariate and goes beyond a restrictive linear regression framework, while the additive structure among covariates remains simple. This balance between non-parametric modeling and a simple additive link facilitates the interpretation of the contribution of each explanatory variable. Still, other regression techniques (neural networks, random forest and other ML algorithms) could replace our GAM approach in the scheme displayed in Figure 3. Keeping in mind that our training period can be short (a few decades) at some locations, this lack of a large training dataset may also limit the application of fully data-driven machine learning techniques.

The model covariates include the following three climatological drivers: (1) smoothed daily temperatures, (2) smoothed daily precipitation, and (3) day of the year (seasonality), and interpolated daily glacier mass balance changes (Figure 3a; for details on datasets see 2.1). The last variable takes into account non-stationarities induced by changing glacier melt. Discharges during the natural and regulated period can have different magnitudes as a result of water diversions, e.g. in the case of hydropower production. Therefore, we normalize both the natural and regulated streamflow time series by dividing by the mean flow over the corresponding period. Such normalization makes natural and regulated flow magnitudes comparable. All other variables were used on their original scales. We use these covariates to fit a GAM for the prediction of streamflow under natural flow conditions (Figure 3b). To do so, we fit the GAM to the streamflow observations of the pre-dam period. As positive and skewed random variables, it is unlikely that streamflow time series follow a Gaussian distribution given the four covariates. To handle

this issue, we choose a Gamma family within the GAM approach and study the following additive link:

$$f_1(p_t) + f_2(h_t) + f_3(d_t) + f_4(g_t), \tag{2}$$

where $p_t$ corresponds to smoothed precipitation, $h_t$ to smoothed temperature, $d_t$ to the day of the year, and $g_t$ to the interpolated glacier mass balance changes (for the implementation, we used the R-package mgcv (Simon Wood, 2022; Wood, 2017)). We assess the model's performance by comparing observed with predicted streamflow values and by computing a range of different performance metrics including the Kling–Gupta (KGE) and Nash–Sutcliffe efficiencies (NSE) (Gupta et al., 2009; Nash and Sutcliffe, 1970), volumetric efficiency (VE, Criss and Winston, 2008), mean absolute error (MAE), root mean squared error (RMSE), and percent bias (PB). The model captures the observed values and their distribution quite well, as illustrated by comparisons of observed vs. predicted values, observed and predicted quantiles, and observed and predicted time series (for an example catchment see Figure 4). This visual impression is confirmed by the goodness-of-fit statistics computed across all 74 catchments (Table 1). KGE values range between a first quartile of 0.38 and a third quartile of 0.75, NSE values between 0.23 and 0.64, and volumetric efficiencies between 0.49 and 0.73, which means that mean flows and volumes are slightly better simulated than high-flows. MAEs range between 0.27 and 0.51 mm/d (normalized flow), the RMSEs between 0.45 and 0.96, and the percentage bias is 0. This performance assessment suggests that the model is suitable for predicting streamflow under natural conditions. Model performance is independent of the length of the record available to fit the GAM but depends on catchment area and elevation (Figure B1). The best performance is achieved in large and high-elevation catchments, while performance is worst in small and low-elevation catchments.

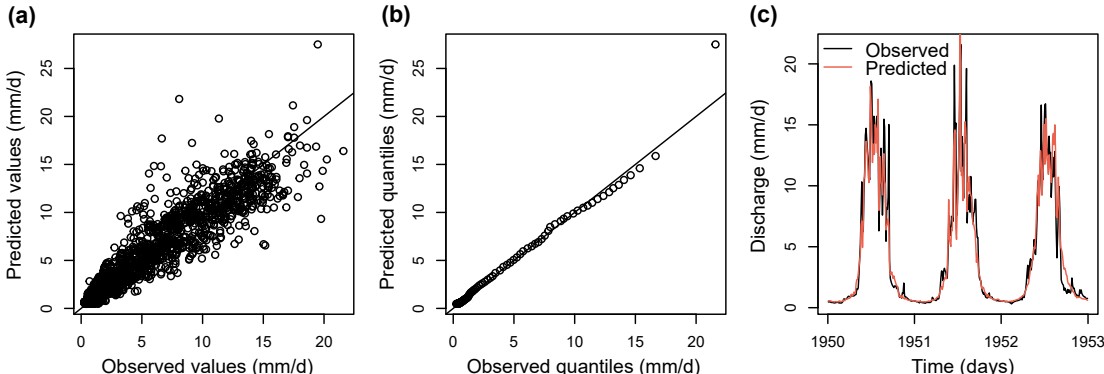

**Figure 4.** Evaluation of the GAM model fitted using natural streamflow data of the Drance de Bagnes (before reservoir construction 1911–1956) and used to predict streamflow with precipitation, temperature, day of year, and glacier mass balance changes as predictors. (a) Observed vs. predicted values (1911–1956), (b) Q-Q plot, observed vs. simulated quantiles (1911–1956), and (c) observed vs. predicted time series (3-years 1950–1953).

Next, we apply this model to deduce the never-observed "natural" flow after the reservoir construction. In this case, the GAM inputs are the same four covariables: temperature, precipitation, day of year, and glacier mass balance changes, but taken over the period after the reservoir construction. As an application example, Figure 8 compares the natural streamflow regime (i.e.

**Table 1.** Performance of GAM in predicting natural streamflow for the pre-regulation period across catchments quantified by different goodness-of-fit statistics including the Nash–Sutcliffe efficiency (NSE, values between 0–1 with an optimum at 1), Kling–Gupta efficiency (KGE, values between 0–1 with an optimum at 1), volumetric efficiency (VE, values between 0–1 with an optimum at 1), mean absolute error (MAE, mm/d), root mean squared error (RMSE, mm/d), and percentage bias (PB, %).

| Statistic | 1st quartile | Median | Mean | 3rd quartile |
|-----------|--------------|--------|------|--------------|
| KGE | 0.38 | 0.48 | 0.53 | 0.75 |
| NSE | 0.23 | 0.31 | 0.34 | 0.64 |
| VE | 0.49 | 0.56 | 0.58 | 0.73 |
| MAE | 0.27 | 0.44 | 0.42 | 0.51 |
| RMSE | 0.45 | 0.77 | 0.84 | 0.96 |
| PB | 0 | 0 | 0 | 0 |

the mean annual hydrograph) of the Drance de Bagnes derived using the model for the regulated period (red) with the natural observed (grey) and the regulated observed streamflow regimes (black). The observed regulated regime has a seasonality distinct from the simulated natural regime. We assume that the difference between the observed regulated streamflow signal and the predicted natural baseline represents the reservoir operation signal.

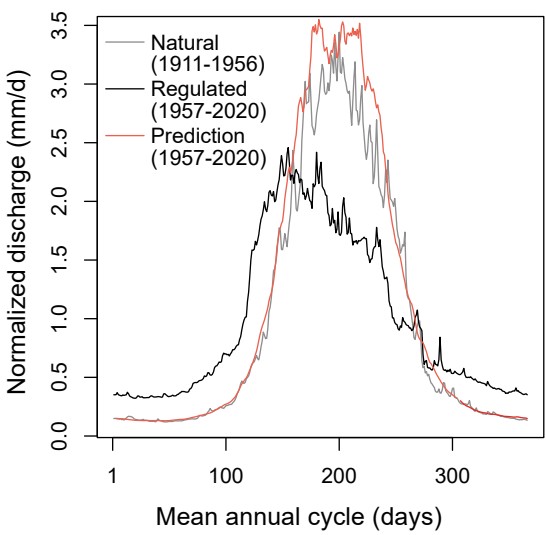

**Figure 5.** Comparison of the observed natural streamflow regime (i.e. the mean annual hydrograph) of the Drance de Bagnes before reservoir construction (grey, 1911–1956), observed regulated regime after reservoir construction (black, 1957–2020), and simulated natural regime for period after reservoir construction (red, 1957–2020).

Under this assumption, we derive the reservoir operation signal by subtracting the predicted 'natural signal' from the observed regulated signal (Figure 6a). To remove noise and retrieve a clear signal, we smooth the signal using regression splines

(Figure 6b). Positive values represent release conditions as the observed regulated signal is higher than the predicted natural signal, while negative values represent storage conditions as the predicted natural signal would be higher than the actually observed regulated signal. The reconstructed signal informs about regulation at a daily scale but can also be aggregated to mean daily values to represent regulation seasonality, i.e. the regulation regime. We here derive reservoir regulation seasonality by averaging the reconstructed daily signals for each day of the year (Figure 6c).

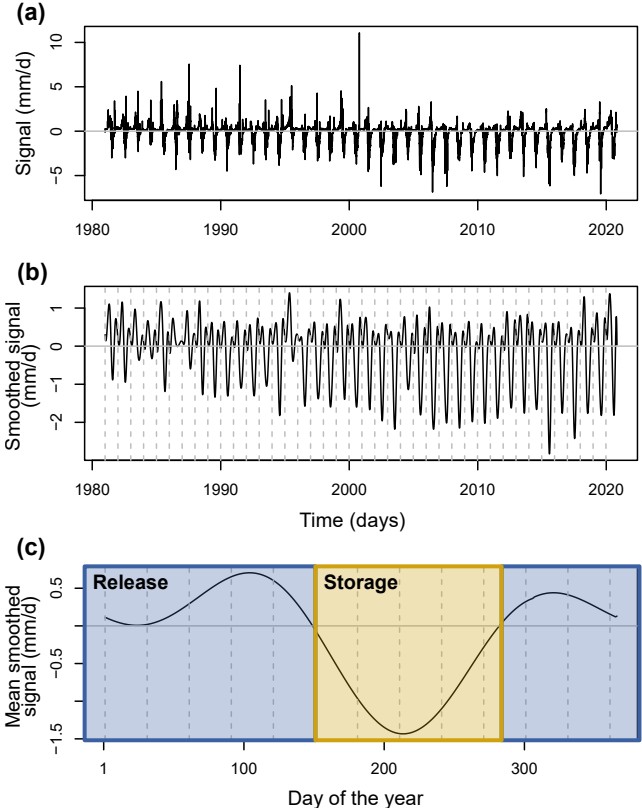

**Figure 6.** Reservoir signal for the Drance de Bagnes reconstructed for the period 1960–2020 using the GAM predictions by subtracting predicted natural discharge from observed regulated discharge, where positive and negative values indicate release and storage, respectively: (a) Raw daily signal, (b) smoothed signal (spline smoothing), and (c) mean seasonal signal.

A direct evaluation of the extracted seasonal reservoir signals is unfortunately not possible because observed inflow and outflow data are not publicly available. Therefore, we evaluated the approach using an alternative validation strategy. The Swiss Federal Office of Energy provides weekly reservoir storage estimates aggregated over a larger region (i.e. canton) (Bundesamt für Energie BFE, 2022). We use these regional storage estimates to compute seasonal changes in regional storage. We then use the regional storage change curve derived for the regions Valais, Grisons, and Ticino to evaluate the reservoir signals extracted using the GAM for all catchments located in the three regions (Figure 7). That is, we apply the GAM approach to the catchments located in the cantons Valais, Grisons, and Ticino using temperature, precipitation, and glacier mass balance changes as covariables. We then compare the extracted reservoir regulation signals to the reservoir signals extracted from the

 regional storage curves. The regulation signal estimates for the ten catchments using the GAM approach compare very well to the signals derived from observed regional reservoir storage data.

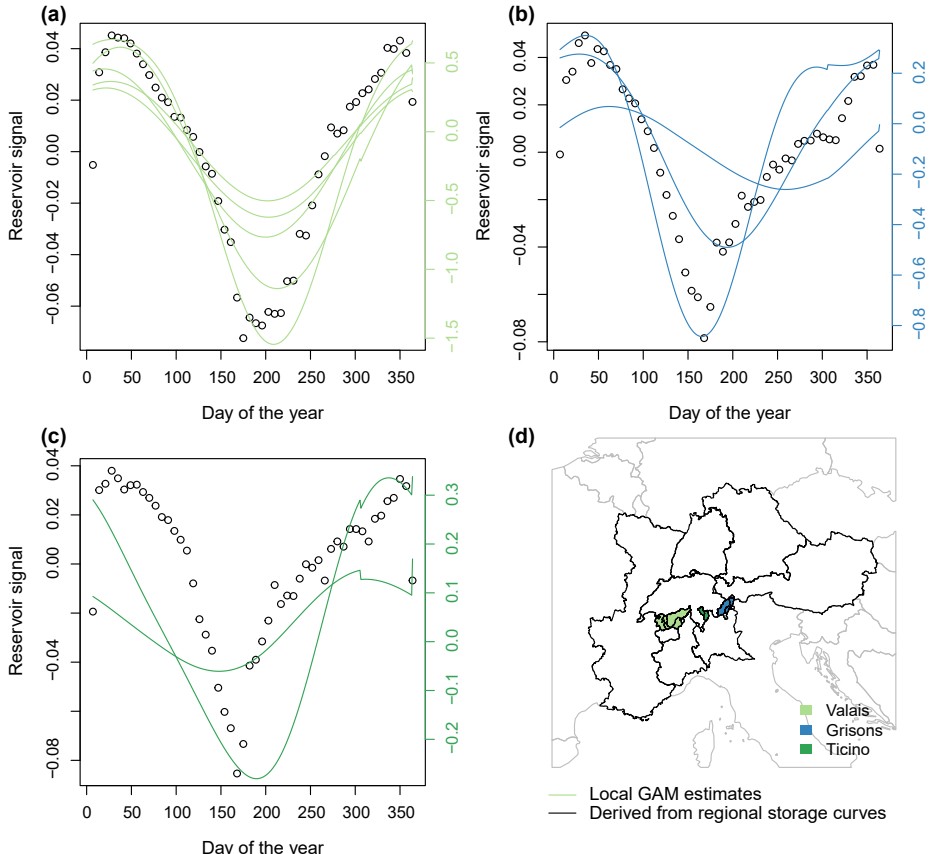

**Figure 7.** Regional reservoir storage change curves for the regions (a) Valais, (b) Grisons, and (c) Ticino derived from regional reservoir storage data provided by the Swiss Federal Office of Energy compared to the reservoir regulation signals estimated using the GAM of the catchments located in the corresponding cantons (d): (a) Rhône, Porte-du-Scex; Rhône, Sion; Rhône, Branson; Drance de Bagnes, Le Châble; and Vispa, Visp, (b) Inn, Tarasp; Inn, Martina; and Spöl, Punt dal Gall, and (c) Brenno, Loderio and Moesa, Lumino.

## 2.3 Reservoir signal variation analysis

We apply the GAM modelling approach introduced in the section above to reconstruct the mean reservoir signals (i.e. reservoir seasonality) of 74 catchments in the Central Alps with streamflow data for a period before and after reservoir construction. We then use these reconstructed reservoir seasonalities to identify groups of catchments with similar reservoir operation patterns using functional data clustering (Ramsay and Silverman, 2002) (Figure 3c). To do so, we follow the approach proposed by Brunner et al. (2020) to cluster streamflow regimes, i.e. mean annual streamflow hydrographs. First, we project the discrete observations, i.e. the reconstructed reservoir operation seasonalities at daily resolution, to a set of B-spline basis functions (R-package fda; Ramsay et al., 2014). B-spline functions are defined by their order of polynomial segments and the amount

of knots, which determine their ability to represent sharp features in a curve (Höllig and Hörner, 2013). Similar to Brunner et al. (2020), we use five spline basis functions of order four, which corresponds to a minimal number of basis functions still allowing for sufficient flexibility in representing diverse shapes of reservoir operation seasonalities. The projection of the observed reservoir operation seasonalities to the five basis functions results in five coefficients per observed operation signal, one per spline base. The analysis is performed in R using the packages *fda.usc* (Febrero-Bande and Oviedo de la Fuente, 2012) and *fda* (Ramsay et al., 2014). Second, we compute a Euclidean distance matrix using the matrix of $n = 74 \times 5$ spline coefficients. Third, we use a hierarchical clustering algorithm (*hclust*) with Ward's minimum variance criterion, which minimizes the total within-cluster variance (Ward, 1963). We cut the tree at $k = 2$ clusters, because this seems to be the most suitable number of clusters given the symmetry of the tree. After cluster identification, we assess various properties of the catchments in the different clusters including the natural streamflow regime, catchment area, and catchment elevation.

## 3   Results of the reservoir signal variation analysis

Reservoir operation in the Central Alps varies by season and across catchments and the types of reservoir signals extracted using the GAM-approach can be grouped into two classes (Figure 8a,b and 9a). In catchments belonging to cluster 1, seasonal flow redistribution from summer to winter is much more pronounced than in catchments belonging to cluster 2. This seasonal redistribution pattern seems to be related to the natural flow regime, which has a more pronounced seasonality in catchments belonging to cluster 1 than those belonging to cluster 2 (Figure 8c,d). The catchments with stronger seasonal redistribution are located at higher elevations and have larger storage capacities than catchments with weaker seasonal redistribution (Figure 9c,d) but do not differ in terms of catchment area (Figure 9a). While some catchments are strongly regulated (i.e. those with strong signal amplitudes), less water is stored and released in other catchments (i.e. those with weak amplitudes) (Figure8a,b). Independent of magnitude, the seasonal release-storage signal appears to be similar in most catchments. Water is mostly stored in summer (negative values), when snowmelt, precipitation, and runoff are abundant (Frei and Schär, 1998; Brunner et al., 2019b; Vorkauf et al., 2021), and released in winter (positive values) when electricity demand is high because of cold temperatures and elevated heating needs (Thornton et al., 2016; Wenz et al., 2017). This regulation seasonality is particularly pronounced in the catchments in the Central Alps, which are identified as a first cluster of catchments sharing similar reservoir operation patterns (Figures 8a and 9a). In this region, reservoirs are mostly operated for hydropower production (Figure 10; Panduri and Hertach (2013) and Brunner et al. (2019a)). In contrast, reservoir operation seasonality is weaker in the catchments in the pre-Alps and lowland areas (Figures 8b and 9c), the second cluster of catchments with similar reservoir operation signals. In this region, reservoirs are operated for a wider variety of purposes including flood protection, recreation, energy production, water and industrial supply (Figure 10; Speckhann et al. (2021)).

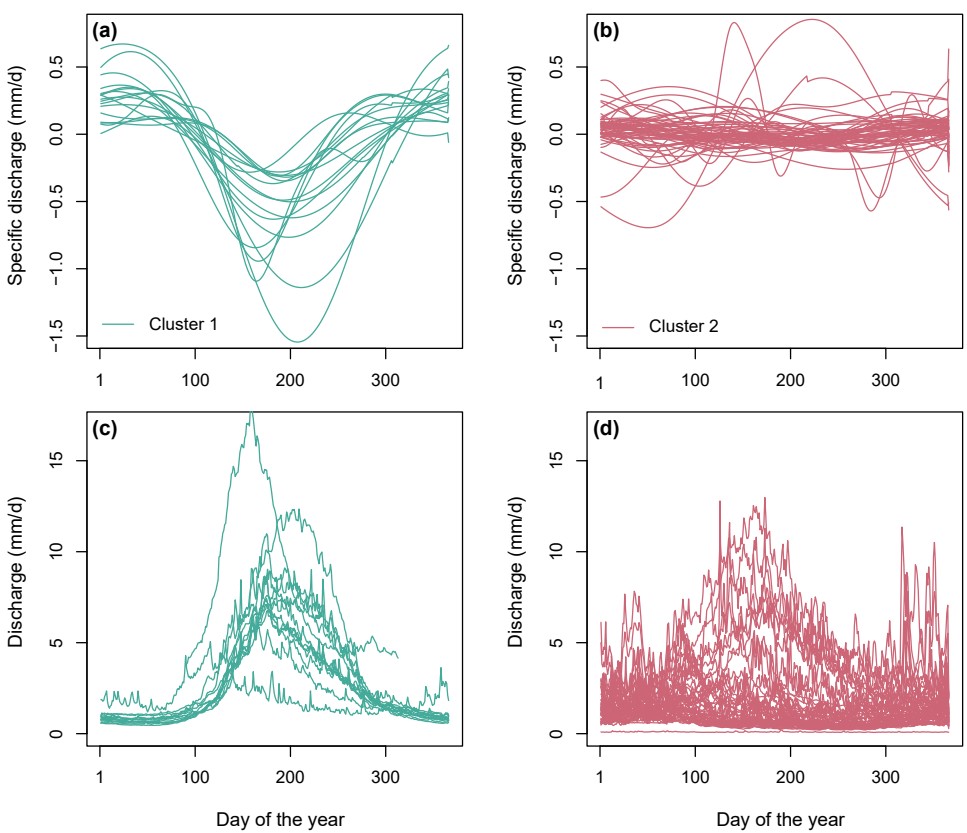

**Figure 8.** Reservoir regulation seasonality patterns clustered into two groups: (a) release in winter and storage in summer and (b) weak seasonal storage pattern. Natural streamflow regimes (computed using the undisturbed streamflow time series before reservoir construction) belonging to the two reservoir regulation clusters (c,d).

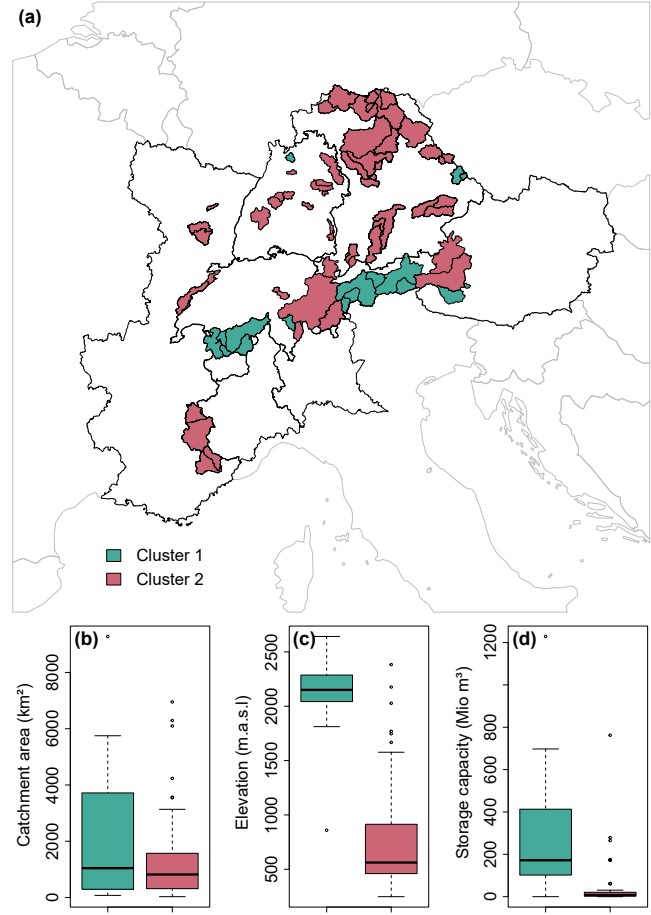

**Figure 9.** Two groups of catchments with distinct reservoir regulation signals: (a) Catchments belonging to cluster 1 (turquoise) and 2 (red) with similar seasonal regulation patterns (see Figure 8), (b) distributions of catchment areas for clusters 1 and 2, (c) distributions of elevations for clusters 1 and 2, and (d) distributions of reservoir storage capacities for clusters 1 and 2.

The catchments belonging to the two clusters clearly differ by elevation and to a weaker degree in catchment area (Figure 9). That is, high-elevation catchments with melt-dominated streamflow regimes show much stronger regulation signals than low-elevation catchments with rainfall-dominated streamflow regimes (Figure 8c,d).

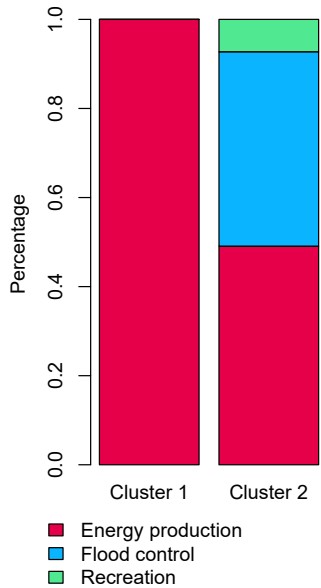

**Figure 10.** Reservoir purpose mix of catchments in regulation clusters 1 and 2 (see Figure 9): Energy production, flood control, and recreational use.

## 4  Discussion

We proposed a generalized additive modelling approach to reconstruct the seasonality and magnitude of reservoir operation using observed streamflow time series, including a period before and after reservoir construction. This statistical approach has the advantage of being observation-based and computationally inexpensive. It does not require setting up a hydrological model to simulate natural streamflow. However, the approach also has some limitations. First, it is only applicable in catchments where streamflow observations are available for a natural period before and a regulated period after reservoir construction. This means that the approach is not applicable in ungauged catchments and in catchments where streamflow is only available for a post-reservoir construction period. Turner et al. (2021) proposed a regionalization approach for reservoir operation signals. Our signals may also be regionalized by establishing a relationship between group membership and catchment characteristics, e.g. elevation, which seems to be strongly related to the type of reservoir regulation signal observed (Figure 8). Second, while the predictive performance of the GAM is satisfactory, there is room for improvement with respect to the simulation of extreme events, which are as in other approaches not perfectly represented. The residuals not only represent the reservoir operation signal, but also include residual model error. Nonetheless, by smoothing the residuals, we are able to reconstruct a regular pattern representing reservoir regulation. As an alternative to GAMs, we tested the use of Generalized Additive Models for Location, Scale and Shape (GAMlss) said to be more appropriate for modelling time series following extreme value distributions. However, such model adaptation did not improve model performance and new statistical modelling frameworks are needed to better represent extreme events. Third, separating flow changes induced by reservoir operation and other types of changes induced by climate change, such as glacier melt contributions, is challenging. While the GAM representing natural

conditions can theoretically consider changes in glacier melt contributions by including glacier mass balance changes, these effects are in practice not perfectly represented because glacier mass balance changes are observed and simulated at a coarse resolution (annual). This means that the signal reconstructed by comparing the simulated natural signal with the observed regulated signal may not solely represent reservoir operation, but to some degree also changes in glacier melt contributions not accounted for by the model. A better separation of the confounding changes – glacier melt and reservoir operation – may be
achieved if more detailed information about glacier mass balance were available or in cases where the seasonality of reservoir regulation is clearly different from the seasonality of glacier melt.

The GAM-approach proposed here can be used to reconstruct reservoir operation signals in other parts of the world, given that streamflow and climate data are available for a period before and after a known date of reservoir construction. Depending on the hydro-climate, the type of predictors used in the GAM might need to be adjusted. For example, the glacier melt part can
be removed in non-alpine regions where streamflow is uninfluenced by glacier melt. The GAM modelling approach introduced here can also be used to assess changes in reservoir operation over time. Such adaptation in reservoir operation might be necessary to account for changing environmental conditions (Feng et al., 2017).

By applying our GAM model to 74 regulated catchments in the Central Alps, we are able to show how reservoir regulation seasonality varies in space. We identify two main groups of regulated catchments (Figure 9): those in the Central Alps with
265 storage in summer and release in winter and those in the pre-Alps and lowland regions with a less pronounced operation seasonality and generally weaker storage and release cycles (Figure 8). The catchments with pronounced regulation cycles in group 1 are mainly operated for hydropower production (Figure 10), while those with less pronounced regulation seasonality in group 2 are operated for a variety of purposes such as flood control or recreation (Figure 10). This finding that lowland catchments have weak reservoir regulation seasonality is in line with findings by Eisele et al. (2004) who have shown that
reservoir regulations in Baden-Württemberg have a very small impact on the timing of hydrological extremes. Applied at a larger or even global scale, the GAM approach could help us to even better understand spatial variations in reservoir operation.

## 5  Conclusions

We develop a generalized additive modelling approach using climate variables as predictors to extract reservoir operation signals from observed streamflow time series available for a period before and after reservoir construction. We apply this approach
to a set of 74 regulated catchments in the Alps to extract reservoir regulation signals at daily resolution by comparing simulated natural flow with observed regulated flow. The mean reservoir operation seasonalities derived from these daily signals are grouped using functional data clustering to identify groups of catchments with similar reservoir operation strategies. We find that in the Central Alps there are two groups of catchments with distinct reservoir operation strategies: high-elevation catchments with pronounced seasonal water redistribution from summer to winter for hydropower production and low-elevation
catchments with weak seasonal water redistribution for different reservoir purposes. The reservoir signals reconstructed using the GAM modelling approach may be used to inform hydrological model development and calibration. Furthermore, the reconstructed signals could inform the representation of reservoir operation in hydrological models. Improving such representation

is crucial to advance the field of change attribution as it will allow for a better separation of climate and regulation signals, which both influence streamflow characteristics.

*Data availability.* data used for our analysis will be published on HydroShare upon acceptance of this manuscript.

## Appendix A: Catchments

**Table A1.** Catchment characteristics of the 74 Alpine catchments used in the analysis: country, name of river, location of gauging station, record length (years), catchment area (km$^2$), elevation (m.a.s.l.), and start year of reservoir operation.

| Country | River | Station | Record length | Start date of record | Catchment area | Elevation | Start year of reservoir operation |
|---|---|---|---|---|---|---|---|
| CH | Rhône | Porte du Scex | 115 | 1905 | 5224 | 2124 | 1957 |
| CH | Rhône | Sion | 104 | 1916 | 3363 | 2287 | 1957 |
| CH | Rhône | Branson | 79 | 1941 | 3718 | 2231 | 1957 |
| CH | Inn | Martina | 116 | 1904 | 1936 | 2342 | 1968 |
| CH | Muota | Ingenbohl | 103 | 1917 | 316 | 1363 | 1966 |
| CH | Brenno | Loderio | 116 | 1904 | 399 | 1812 | 1963 |
| CH | Drance de Bagnes | Le Châble, Villette | 109 | 1911 | 253 | 2601 | 1956 |
| CH | Doubs | Ocourt | 99 | 1921 | 1272 | 960 | 1953 |
| CH | Spöl | Punt dal Gall | 69 | 1951 | 294 | 2390 | 1968 |
| CH | Inn | Tarasp | 63 | 1957 | 1577 | 2383 | 1968 |
| CH | Doubs | Combe des Sarrasins | 67 | 1949 | 996 | 985 | 1953 |
| CH | Vispa | Visp | 117 | 1903 | 784 | 2642 | 1965 |
| CH | Moesa | Lumino, Sassello | 108 | 1912 | 471 | 1667 | 1958 |
| AT | Rhein | Lustenau (Höchster Brücke) | 66 | 1951 | 6289 | 1770 | 1976 |
| AT | Bregenzerach | Kennelbach | 66 | 1951 | 826 | 1120 | 1979 |
| AT | Vils | Vils (Lände) | 56 | 1961 | 198 | 1274 | 1965 |
| AT | Inn | Prutz | 66 | 1951 | 2454 | 2284 | 1966 |
| AT | Inn | Magerbach | 66 | 1951 | 5091 | 2212 | 1966 |
| AT | Inn | Innsbruck (oberh. Sill) | 66 | 1951 | 5750 | 2139 | 1981 |
| AT | Ziller | Zell am Ziller-Zellbergeben | 66 | 1951 | 695 | 2056 | 1969 |
| AT | Inn | Kirchbichl - Bichlwang | 66 | 1951 | 9279 | 1941 | 1986 |
| AT | Salzach | Golling | 66 | 1951 | 3547 | 1577 | 1958 |
| AT | Salzach | Oberndorf | 56 | 1961 | 6099 | 1340 | 1974 |
| AT | Mur | Muhr | 56 | 1961 | 76 | 2043 | 1991 |
| AT | Möll | Kolbnitz a. d. Tauernbahn VHP | 46 | 1971 | 1045 | 1935 | 1981 |
| FR | La Moselle | Libellé station | 50 | 1970 | 627 | 724 | 1983 |
| FR | La Moselle | Libellé station | 50 | 1970 | 1214 | 653 | 1983 |
| FR | La Plaine | Libellé station | 50 | 1970 | 117 | 514 | 1986 |
| FR | La Durance | La Durance à Briançon [aval] | 62 | 1955 | 202 | 2150 | 2000 |
| FR | La Durance | La Durance à l' Argentière-la-Bessée | 111 | 1910 | 961 | 2177 | 1966 |
| FR | La Durance | La Durance à Espinasses [Serre-Ponçon] | 69 | 1948 | 3567 | 2028 | 1966 |
| FR | La Tinée | La Tinée à la Tour [Pont de La Lune] | 44 | 1972 | 703 | 1746 | 2006 |
| FR | Le Var | Le Var à Malaussène [La Mescla] | 51 | 1965 | 1824 | 1482 | 2006 |
| DE | Baierzer Rot | Achstetten | 96 | 1924 | 264 | 631 | 1971 |
| DE | Jagst | Schwabsberg | 79 | 1941 | 178 | 514 | 1968 |
| DE | Würm | Schafhausen | 68 | 1952 | 237 | 492 | 1976 |
| DE | Rot | Binnrot | 60 | 1960 | 130 | 679 | 1971 |
| DE | Nagold | Nagold | 52 | 1941 | 376 | 625 | 1965 |
| DE | Kinzig | Schwaibach | 106 | 1914 | 952 | 604 | 1978 |
| DE | Erms | Riederich | 98 | 1922 | 159 | 637 | 1962 |
| DE | Rems | Schorndorf | 89 | 1931 | 417 | 432 | 2006 |
| DE | Zaber | Hausen | 89 | 1931 | 108 | 257 | 1968 |
| DE | Schwarzbach | Eschelbronn | 64 | 1954 | 191 | 248 | 2000 |
| DE | Lauter | Süßen | 79 | 1941 | 68 | 562 | 1976 |
| DE | Lein | Abtsgmünd | 99 | 1921 | 246 | 492 | 1957 |
| DE | Jagst | Dörzbach | 97 | 1923 | 1027 | 458 | 1958 |
| DE | Nagold | Calw | 79 | 1941 | 586 | 603 | 1965 |
| DE | Rottach | Greifenmühle | 63 | 1957 | 31 | 914 | 1984 |
| DE | Wertach | Biessenhofen | 100 | 1920 | 444 | 884 | 1959 |
| DE | Altmühl | Treuchtlingen | 91 | 1929 | 990 | 469 | 1974 |
| DE | Altmühl | Eichstätt | 91 | 1929 | 1391 | 482 | 1974 |
| DE | Naab | Unterköblitz | 80 | 1940 | 2002 | 514 | 1965 |
| DE | Schwarzach | Warnbach | 80 | 1940 | 819 | 551 | 1960 |
| DE | Schwarzer Regen | Teisnach Schwarzer Regen | 90 | 1930 | 624 | 782 | 1976 |
| DE | Kleiner Regen | Lohmannmühle | 59 | 1961 | 116 | 859 | 1976 |
| DE | Chamb | Furth im Wald | 70 | 1950 | 279 | 540 | 1989 |
| DE | Chamb | Kothmaißling | 60 | 1960 | 408 | 522 | 1989 |
| DE | Amper | Fürstenfeldbruck | 100 | 1920 | 1248 | 744 | 1961 |
| DE | Amper | Inkofen | 95 | 1925 | 3135 | 622 | 1961 |
| DE | Maisach | Bergkirchen | 85 | 1935 | 1581 | 705 | 1961 |
| DE | Vils | Rottersdorf | 81 | 1939 | 722 | 475 | 1972 |
| DE | Vils | Grafenmühle | 81 | 1939 | 1433 | 443 | 1972 |
| DE | Rott | Birnbach | 90 | 1930 | 854 | 460 | 1960 |
| DE | Main | Schwürbitz | 80 | 1940 | 2414 | 488 | 1968 |
| DE | Main | Kemmern | 90 | 1930 | 4235 | 434 | 1968 |
| DE | Rodach | Unterlangenstadt | 90 | 1930 | 712 | 530 | 1968 |
| DE | Itz | Coburg | 95 | 1925 | 363 | 458 | 1982 |
| DE | Itz | Schenkenau | 53 | 1967 | 505 | 423 | 1982 |
| DE | Regnitz | Pettstadt | 98 | 1922 | 6951 | 404 | 1956 |
| DE | Rednitz | Neumühle Rednitz | 110 | 1910 | 1816 | 424 | 1975 |
| DE | Roth | Roth Bleiche | 52 | 1968 | 179 | 414 | 1985 |
| DE | Pegnitz | Nürnberg Lederersteg | 110 | 1910 | 1180 | 457 | 1956 |
| DE | Fränkische Saale | Bad Kissingen Golfplatz | 91 | 1929 | 1572 | 382 | 1965 |
| DE | Fränkische Saale | Wolfsmünster | 90 | 1930 | 2116 | 374 | 1965 |

## Appendix B: Further model evaluation

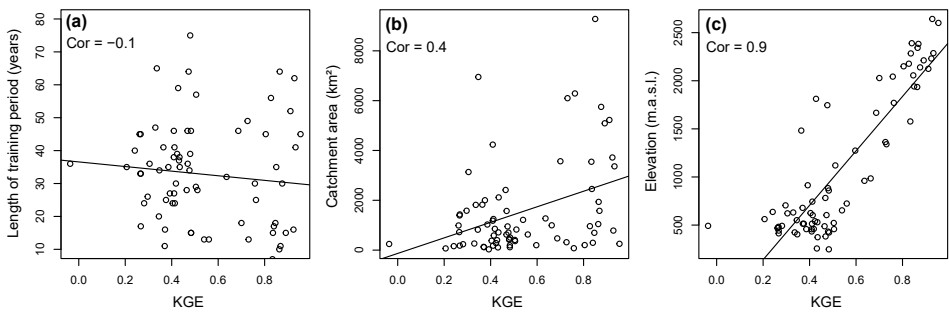

**Figure B1.** Relationship between model performance and catchment characteristics: (a) record length used to fit the GAM, (b) catchment area, and (c) elevation.

*Author contributions.* MIB developed the concept and jointly with PN the methodology of this study. MIB performed all analyses, produced the figures, and wrote the first draft of the manuscript, which was revised and edited by PN.

*Competing interests.* The authors declare no competing interests. MIB is an editor with HESS.

*Acknowledgements.* We thank the German Research Foundation (grant 2100371301), the French national programs (FRAISE- LEFE/INSU and 80 PRIME CNRS-INSU), and the European H2020 XAIDA (Grant agreement ID: 101003469) for supporting this research. We acknowledge the E-OBS dataset from the EU-FP6 project UERRA (https://www.uerra.eu) and the Copernicus Climate Change Service, and the data providers in the ECA&D project (https://www.ecad.eu). Philippe Naveau also acknowledges the support of the French Agence Nationale
de la Recherche (ANR) under reference ANR-20-CE40-0025-01 (T-REX project), and the ANR-Melody (ANR-19-CE46-0011). We thank Loris Compagno for providing the simulated glacier mass balance time series. In addition, we thank the two anonymous reviewers for their constructive feedback, which helped to greatly improve the presentation of the results.

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
