# Peer review of "Spatial variability in Alpine reservoir regulation: deriving reservoir operations from streamflow using GAMs"

_Hydrology and Earth System Sciences, 2022_

## Author Comment (AC1)

**Reviewer 1**

This paper uses generalized additive models (GAMs) to compare pre- and post-dam construction flows as a function of rainfall, temperature and glacier mass changes in 74 high mountain catchments. The approach enables to reconstruct reservoir operations, as well as its hydro-climatic drivers, in situation where inflows and outflows are not readily available (which is often the case). The paper is really interesting, the method is a smart way to tackle a challenging puzzle, and there is a feeling that it is likely novel… but this is incumbent on authors to demonstrate, both by explaining the novelty in more detail, and by validating the method! My main remarks are as follows:
**Reply:** *Thank you very much for this valuable and constructive feedback, which we highly appreciate. Please find our responses to the detailed comments below.*

1) the novelty needs to be explicitly and precisely stated on the introduction.
**Reply:** *Thank you very much for highlighting the need to more specifically describe the novelty of our study. The novelty is twofold, as specified in the introduction. First, we 'propose a statistical three-step approach for reservoir signal reconstruction in catchments where reservoir outflow but no inflow time series are available'. Second, we 'shed light on spatial variations in reservoir regulation signals and their relationship to catchment characteristics'. 'The combination of the proposed reservoir signal reconstruction approach with functional clustering allows us to provide insights into how reservoir regulation varies spatially in the Alps and to which degree these variations are related to catchment characteristics.'*

2) the absence of a formal validation is a real issue that should (and can!) be addressed.
**Reply:** *Thank you for pointing out the need of a formal validation. We expanded both the validation of the GAM model and compared the extracted reservoir regulation signals to an alternative signal derived from regional reservoir storage curves provided by the Swiss Federal Office of Energy (for more detail see our response below).*

3) the methodology is creative, but the explanations are sometimes vague, and needs to be better justified and documented in several places.
**Reply:** *Thank you very much for indicating the need for clarification. We carefully revised and expanded the methods section, by (1) adding an illustration of the workflow (Figure 3); (2) moving the technical information from the introduction to the methods section; and (3) providing additional details on the different working steps, in particular model evaluation.*

4) results need to be presented step by step. They are really interesting, but authors need to be more rigorous in presenting how they got to them.
**Reply:** *We present the results related to model evaluation in the Methods section together with detailed figure descriptions, while we present the results related to the reservoir signal variation analysis in the results section. We consolidated the results section by redesigning Figures 7 and 8 following recommendations by Reviewer 2.*

It might take a substantial revision for authors to address the comments above. Revisions are important because this work has the potential to be a nice paper. I now develop the four points above.

1) Several points need to be clarified in the introduction:

=> What is the exact scope of the paper? Is the type of data authors base themselves on widely available in a range of cases (not just the Alps)?

**Reply:** *Thank you for highlighting the need to better describe the scope of the paper. We specified in the introduction that: 'This approach can be used to reconstruct reservoir operation signals in catchments where streamflow and climate data are available for a period before and after a known date of reservoir construction. Such information is more widely available than reservoir in- and outflow measurements, which means that the approach is applicable in different regions around the globe where streamflow observations and information on reservoir construction dates are available. Here, we apply the approach to extract reservoir signals from observed time series of 74 catchments in the Central Alps.'*

=> Why are GAM the appropriate methodology here? Why not use machine learning methods for instance?

**Reply:** *We agree that other methods could potentially be used for predicting natural streamflow. We aimed to identify a relatively simple and parsimonious model using a small number of explanatory variables. Using machine learning approaches would only make sense if many different explanatory variables were used for model fitting, which is not the case here. We added a short discussion of the advantages of GAMs compared to simpler and more complex model alternatives to the introduction: 'The main advantage of GAMs is that cubic spline modeling offers flexibility for each covariate and goes beyond a restrictive linear regression framework, while the additive structure among covariates remains simple. This balance between non-parametric modeling and a simple additive link facilitates the interpretation of the contribution of each explanatory variable. Still, other regressions techniques (neural networks, random forest and other ML algorithms) could replace our GAM approach in the scheme displayed in Figure 3. Keeping in mind that our training period can be short (a few decades) at some locations, this lack of a large training dataset may also limit the application of fully data-driven machine learning techniques.'*

=> A simple search of "generalized additive model reservoir" on Scopus returns 120 results. Can authors ascertain no similar attempt has been tried in the past? What have been GAM uses in the literature around hydrological and reservoir modelling?

**Reply:** *GAMs are versatile and flexible models for a range of prediction applications and have been previously used in the context of reservoirs, e.g. to predict reservoir sedimentation loads or reservoir water quality. We acknowledged these alternative uses of GAMs in the context of reservoirs in the introduction: 'In this study, the response variable $y_t$ corresponds to streamflow time series in mm/d (units). Alternatively, GAMs have in the context of reservoirs also been used to predict other variables than streamflow such eutrophication levels (Catherine et al. 2010) or downstream water temperatures (Coleman et al. 2021). ' However, we are unaware of any study that has used GAMs to infer the reservoir operation signal from observed streamflow time series covering a period before and after reservoir construction.*

2) Because there is no inflow and outflow data for the catchment authors study, they do not validate the approach using basins for which the post-dam construction reservoir inflows and outflows are known. But there are many mountain reservoirs in other regions (e.g., the USA) for which inflows and outflows, as well as the other data authors use in the Alps. This would provide

formal validation. Note they can then choose to split this paper into two papers (i) one that presents and validates the method, (ii) one that applies it to Alpine catchments to derive new knowledge (which is what they have here). I note here that the confidential dataset shared by authors does not address this concern about validation.

**Reply:** *We agree that some sort of direct validation on observed reservoir storage changes is desirable. Observed inflow and outflow data are hardly available and naturalized inflow is often simulated. Therefore, we looked for an alternative validation strategy. The Swiss Federal Office of Energy provides weekly reservoir storage estimates aggregated over a larger region (i.e. canton). We used these regional storage estimates to compute seasonal changes in regional storage. We then used the regional storage change curve derived for the region Valais to evaluate the reservoir signal extracted using the GAM for a catchment located in that region (see Figure 1). The regulation signal estimated for the Rhône catchment using the GAM approach compares really well to the signal derived from observed regional reservoir storage data. We added this evaluation based on observed data to the manuscript.*

[Figure]

*Figure 1: Observed reservoir storage change curve for the region Valais derived from regional reservoir storage data provided by the Swiss Federal Office of Energy compared to the estimated reservoir regulation signal for the catchment Rhône (Porte-du-Scex), which is located in the Valais region.*

3) In order of appearance, my remarks are as follows:

Performance of the pre-dam model: with 74 catchments, it would only be normal that the GAMs model would work more or less well across the sample. The methodology section (lines 126-130 in particular) does not explain what performance measures are used. Related question: do authors investigate the influence of the length of the pre-development record on model quality? Please include a full list (maybe in supplementary material) of the catchments, including relevant characteristics (e.g., flow record length, year reservoir went online, catchment surface, ratio of reservoir storage over natural annual flow).

**Reply:** *Thank you for highlighting that a broader goodness-of-fit assessment was required. We specified in the methods section that: 'We assess the model's performance by comparing observed with predicted streamflow values and by computing a range of different performance metrics including the Kling-Gupta (KGE) and Nash-Sutcliffe efficiencies (NSE) (Gupta et al. 2009, Nash 1970), volumetric efficiency (Criss et al. 2008), mean absolute error (MAE), root mean squared error (RMSE), and percent bias (PB).' The results of this analysis are presented in the new Table 1. In addition, we added a full list of all catchments including their catchment characteristics to the*

*appendix (Table 1A): Country, river, measurement station, record length, catchment area, mean elevation, and start year of reservoir operation.*

Authors give no indication whether the possibility of an extended filling period for the reservoirs was considered. Or did they assume this would be negligible? If so, please justify explicitly.
**Reply:** *We specified that with 'reservoir construction date', we mean the 'date when the reservoir went into operation.'*

Section 2.2: the method enables one to derive time series of inflows / storage / release. Authors choose a final signal in terms of difference outflows-inflows that is creative and innovative, and could be a key contribution of the paper. Yet it is not justified, especially in relation to the literature on the role of reservoir in hydrological models that authors claim to contribute to. This leads to several important questions:

i) How do the chosen signals relate literature trying to specify reservoir outflows as a function of inflows, storage, and reservoir purpose?
**Reply:** *Thank you for highlighting the need to discuss similarities and differences to other types of signals described in the literature. We specified that: 'This resulting signal indicates how much water is stored in and released from reservoirs in which season (i.e. day of the year). These reservoir-storage-seasonality signals take a reservoir perspective and provide information on storage in addition to releases, but not on inflow. Therefore, they are distinct from the signals extracted through other approaches, e.g. simulated water releases (Coerver et al. 2018); spectral differences between in- and outflows highlighting the time scales most affected by reservoir regulation (White et al. 2005, Shiau et al. 2014); or water storage and release policies, which define release decisions as a percent deviation from long term mean inflow (Turner et al. 2021).'*

ii) Why not derive relationships between inflows and outflows once inflows are derived? That is more standard.
**Reply:** *Thank you for this suggestion. Our approach does not provide information on reservoir inflows. Instead, it provides information on when and how much water is released from and stored in reservoirs. We clarified this in the introduction by writing: 'This resulting signal indicates how much water is stored in and released from reservoirs in which season (i.e. day of the year). These reservoir-storage-seasonality signals take a reservoir perspective and provide information on storage in addition to releases, but not on inflow.'*

iii) How do we know smoothing actually is needed? And how do we know how to parameterise the smoothing technique?
**Reply:** *We compared the performance of the GAM driven with smoothed covariates with the performance of a GAM driven with raw (unsmoothed) covariates. This comparison showed that model performance can be improved by using smoothed instead of raw data, which is why we decided to use the smoothed covariates for our model. A comparison of observed natural flow with simulated flow derived using a GAM with smoothed covariates and a GAM with raw covariates is shown in Figure 2, which indicates that simulated values are closer to the observed values when using the GAM with the smoothed covariates. This visual impression is confirmed when computing and comparing the NSE and KGE performance metrics for the two model types across all 74*

*catchments (Figure 3 in this response to the reviewers). In almost all catchments, the GAM with smoothed covariates performs better both in terms of KGE and NSE.*

[Figure]

*Figure 2: Comparison of simulated and observed natural flow for an example catchment where the simulated flow is derived with a GAM fitted on the smoothed covariates (model used in the study, upper panel) and a GAM fitted on the raw covariates (lower panel).*

[Figure]

*Figure 3: Goodness-of-fit ((a) KGE and (b) NSE) of observed and simulated flow derived using two types of models: (1) GAM with smoothed covariates and (2) GAM with raw covariates.*

4) The results section is very interesting in highlighting a category of high-elevation hydropower reservoirs (but authors should comment in the discussion how new this is). But it only reports on what happens once the flow signal is derived, whereas it would be great to see results for the fitting of the pre-dam GAM model, as this is key to getting quality results. Authors should report in detail on goodness of fit, relationship between number of years of record and quality of the results, etc.
**Reply:** *Thank you for highlighting the need for a more in-depth performance assessment. We provide results of the fitting of the pre-dam ('natural') GAM model in the methods section. In Figure 4, we compare observed natural flows to simulated natural flows of an example catchment. We extended the performance analysis to all catchments by computing a range of different goodness-of-fit statistics (Nash-Sutcliffe efficiency, Kling-Gupta efficiency, percent bias, volumetric efficiency, mean absolute error, and the root mean squared error) for all basins and display them in the new Table 1. We also added a new figure to the appendix showing the relationship of model*

*performance (in terms of KGE) to the length of the record available for model fitting, catchment area, and elevation. We described the results of this additional goodness-of-fit analysis as follows: 'Model performance is independent of the length of the record available to fit the GAM but depends on catchment area and elevation (Figure B1). The best performance is achieved in large and high-elevation catchments, while performance is worst in small and low-elevation catchments.'*

[Figure]

*Figure 4: Kling-Gupta efficiency vs. length of training period, catchment area, and elevation.*

Another remark on results at lines 161-163: this is a key point that seems difficult to make without taking a long look at reservoirs' storage capacities. Reservoir with small regulation capacity (ratio of live storage to average annual flow) will naturally have weaker signals.
**Reply:** *We expanded the description of the results and added a subpanel to the new Figure 8 (d) showing the distribution of reservoir storage capacities across catchments belonging to clusters 1 and 2. The catchments with the larger storage capacities are those with the stronger seasonal redistribution.*

One last remark concerning results is that it would be great to highlight one or two known water supply reservoirs (for irrigation or any other uses), and show how their method enables quantifying downstream needs. That's not strictly necessary but it would show that the proposed method is versatile (which it is), instead of being a one-trick pony that only identified high-elevation single use hydropower reservoir.
**Reply:** *Thank you very much for this suggestion of using the extracted regulation signals to estimate downstream water demand. We think that the reservoir signals cannot be used as direct estimates of water demand because part of the water demand might be covered by other sources of water such as groundwater storage or precipitation.*

Finally, below are also some minor remarks:

Lines 47-51: You talk about a two step setup and then lose the reader a bit by not making both steps explicit. I'd advise doing that.
**Reply:** *Thank you for highlighting the need for clarification. We now talk about a 'three-step' approach and specified that: 'In a first step, the approach fits a GAM to streamflow observations representing natural pre-reservoir conditions using precipitation, temperature, day of the year, and glacier mass balance changes as covariates. In a second step, this GAM is applied to covariates derived for the regulated post-dam period to predict natural streamflow for this regulated period. In a last step, the reservoir regulation signal is reconstructed by subtracting the predicted 'natural signal' from the observed regulated signal.'*

Line 56: by convention, please introduce an equation with ":" Several times in the paper please replace "see, e.g." with "see e.g.,".
**Reply:** *We added ':' before equations and replaced 'see, e.g.' by 'see e.g.'.*

Line 113: please replace "(GAM) (Hastie" by "(GAM; Hastie" Figure 3.c: the x-axis says this is the early 1950s whereas the legend says early 1910s. Which is it? Also, please fit the limits on the x-axis to the beginning and ending of the record you are plotting (also valid in Figure 4).
**Reply:** *We corrected the text formatting and corrected the figure caption.*

Figure 4: please enlarge the figure.
**Reply:** *We enlarged the figure.*

"Glaciermelt": consider writing "glacier melt" instead.
**Reply:** *We replaced 'glaciermelt' with 'glacier melt'.*

Line 203-207: yes, but it would be great to try to nail the scope in the introduction instead of just speculating in the discussion. How applicable is this data to other places? That can be in principle deduced from the methodological setup alone.
**Reply:** *We specified the data-requirements for the application of the approach in the discussion: 'The GAM-approach proposed here can be used to reconstruct reservoir operation signals in other parts of the world, given that streamflow and climate data are available for a period before and after a known date of reservoir construction.' We also stated in the introduction that: 'This approach can be used to reconstruct reservoir operation signals in catchments where streamflow and climate data are available for a period before and after a known date of reservoir construction. Here, we apply the approach to extract reservoir signals from observed time series of 74 catchments in the Central Alps.'*

---

## Author Comment (AC2)

**Reviewer 2**

This paper introduces a methodology and analysis of estimating reservoir operations across 74 different catchments in the Alps using general additive models. The goals were to identify groups of catchments with similar reservoir operations and assess how catchments differ with diff operation differ in location and catchment characteristics (how do different operations affect locations and areas). The approach of using GAMS was indeed a really interesting idea, however, I found the manuscript to be too vague and light on details to be confident in the results or for the methods to be reproduced.

**Reply:** *Thank you very much for your careful assessment of our manuscript and for acknowledging the value of our work. We highly appreciate your constructive inputs. We substantially revised the manuscript with a particular focus on the methods and results sections. In particular, we provided an illustration of the workflow, which allowed us to deepen the methods description, we added a systematic assessment of the model's performance across all catchments, and redesigned the figures in the results section. We hope that the revised methods section provides a clearer idea of how the model was implemented and how it performs and the revised results section is easier to follow.*

**Major Edits by Section:**

Methods:

- General Data Section

  o Can correlations be made if we are not looking at the same time periods? Can this be expanded on?
  **Reply:** *We specified in the introduction that: 'The estimated transfer function calibrated on the unregulated period is applied via equation 1 to the regulated period, assuming that the relationship between climate variables and streamflow remains stable over time.'*

- Climate data was from gridded dataset, averaged over the full time period, moving time window to and replaced NA with mean flow

  o Does it make sense to replace NA with mean flow? If you want to keep mean flow, perhaps filling with monthly averages would be better.
  **Reply:** *Thank you for this suggestion. We replaced the few missing values with daily averages: 'If present, missing values in the time series of all variables are replaced by the daily mean over the natural period for the natural data and over the regulated period for the regulated data.'*

  o How was the disaggregation done?
  **Reply:** *We specified that: 'Each annual mass balance time series is dis-aggregated into daily resolution by smoothing the annual signals over 365 days.'*

  o What was done to normalize or standardize the other data used
  **Reply:** *We specified that 'All other variables were used on their original scales.'*

- GAMs Section

o   I'm not sure subtracting the mean standardizes your streamflow timeseries. Perhaps, it would be better to normalize by subtracting the mean and dividing by the standard deviation
**Reply:** *We agree that normalizing by subtracting the mean and dividing by the standard deviation would be a valid normalization alternative. However, we normalized by dividing by the mean because we wanted to prevent the creation of negative flow values. We updated the text accordingly: 'We normalize both the natural and regulated streamflow time series by dividing by the mean flow over the corresponding period.'*

o   Do you test for outliers in the Streamflow data? The peak in Figure 5 a around 2000 could be a result of an extreme weather event or an outlier in the observed data. What do you do to fix these if they occur, or do you simply assume that by normalizing you remove all the outliers?
**Reply:** *We are using streamflow data that has been quality checked by the national and regional data providers. Therefore, we assume that the outliers in the observed data are related to actual extreme events rather than measurement errors. The normalization does not remove outliers.*

▪   Is it observed – natural for your comparison? If it is not, then the phrasing around line 140 needs to be changed.
**Reply:** *We clarified the sentence by writing: 'Positive values represent storage conditions as the observed regulated signal is higher than the predicted natural signal, while negative values represent storage conditions as the predicted natural signal would be higher than the actually observed regulated signal.'*

- Signal Variation Analysis (2.3)

o   I think this section would benefit from a graphic explaining the workflow or at least demonstrating how the clustering is going as this felt like the most information rich section with a lot of steps.
**Reply:** *Thank you for this great suggestion. We created an illustration depicting the different steps of our workflow (new Figure 3): (a) Input data, (b) GAM modelling and reservoir signal reconstruction, and (c) reservoir signal clustering. Panel (c) illustrates the different steps involved in the functional data clustering procedure: (1) functional data representation and (2) functional data clustering.*

o   I would also add more details on the previous Brunner et al., 2020 paper.
**Reply:** *In addition to adding an illustration of the clustering workflow, we provide some more information on the functional data representation: 'B-spline functions are defined by their order of polynomial segments and the amount of knots, which determine their ability to represent sharp features in a curve (Höllig et al. 2013).*

- My main concern is that figures 3,4,5 only focus on the single catchment and not all the catchments. Would it be possible to include all the catchments.
**Reply:** *Thank you for highlighting that model evaluation had to be expanded beyond example catchments. We added a more comprehensive model evaluation by computing a range of goodness-of-fit statistics for all catchments in the dataset: 'We assess the model's performance by comparing observed with predicted streamflow values and by computing a range of different performance metrics including the Kling-Gupta (KGE) and Nash-Sutcliffe efficiencies (NSE) (Gupta et al. 2009, Nash 1970), volumetric efficiency (VE, Criss & Winston 2008), mean absolute error (MAE), root mean squared error (RMSE), and percent bias (PB).*

*The model captures the observed values and their distribution quite well, as illustrated by comparisons of observed vs. predicted values (panel a), observed and predicted quantiles (panel b), and observed and predicted time series (panel c) (for an example catchment see Figure 4). This visual impression is confirmed by the goodness-of-fit statistics computed across all 74 catchments (Table 1). KGE values range between a first quartile of 0.38 and a third quartile of 0.75, NSE values between 0.23 and 0.64, volumetric efficiencies between 0.49 and 0.73, which means that mean flows and volumes are slightly better simulated than high-flows. MAEs range between 0.27 and 0.51 mm/d (normalized flow), the RMSEs between 0.45 and 0.96, and the percentage bias is 0. This performance assessment suggests that the model is suitable for predicting streamflow under natural conditions.'*

- Figure 1: I would zoom out a bit from the map so we can see the full Rhine and be more oriented in the catchments, Also what are the differences between the purple and black outlines? Denote that in the figure caption
  **Reply:** *Thank you for these valuable suggestions. We adjusted the figure by zooming out and by adding a legend including the different map symbols.*

  o Also, may be useful to put a map of the reservoirs so we can see where they are spatially located
  **Reply:** *We like the idea of adding reservoir locations. However, adding all reservoirs would make the map a bit too noisy and we therefore decided to stay with catchment boundaries and outlets.*

- Figure 5: I would shade the release periods vs storage periods in panel c so that the reader has an easier visualization for what periods are storage and which ones are releases.
  **Reply:** *We shaded the storage and release in Figure 5c (new Figure 6c).*

  Results:

- This section felt very choppy as there was one figure then a few lines of text. I think this could be more clear by grouping the results and creating panel plots. I do like that this section had all the catchments on it
  **Reply:** *Thank you for indicating that this section needed improvement. We substantially reduced the number of figures in the results section by removing Figure 6 and creating two new multi-panel figures. The new Figure 7 combines the results previously shown in Figures 7 and 10 and the new Figure 8 combines the results previously displayed in Figures 8 and 9.*

  o Figure 6: I'm not sure this figure adds too much to the discussion. Perhaps you could panel your reservoirs by bigger basin or by similar characteristics ( ie peaks in summer, peaks in winter). Another option is to color them by region or use, although I do think paneling or grouping would be useful to the viewer.
  **Reply:** *We agree that the previous Figures 6 and 7 provided redundant information and think that it is a good idea to remove one of them. Therefore, we removed Figure 6 and replaced it with a workflow illustration as suggested (new Figure 3).*

  o Also, after seeing Figure 7, I think you can cut figure 6 and use that space to add a graphic about the workflow in 2.3
  **Reply:** *We agree that the previous Figures 6 and 7 provided redundant information and think*

*that it is a good idea to remove one of them. Therefore, we removed Figure 6 and replaced it with a workflow illustration as suggested (new Figure 3).*

- o I would also add a legend to this plot to denote what green and blue mean. Additionally, I would pick more colorblind friendly colors to be more inclusive.
  **Reply:** *We added a legend to the plot and changed the colors to a colorblind-friendly color combination. In addition, we adjusted the cluster colors in all other figures.*

- Figure 8 should have a map of the larger area so we can situate ourselves a little better
  **Reply:** *We adjusted the extent of the figure in order to improve orientation.*

- You could combine figure 9 and 8 into one panel so this section doesn't feel as choppy
  **Reply:** *Thank you for this great suggestion. We combined figures 8 and 9.*

- Figure 10's results tie directly into the map created in Figure 8. I would definitely panel some of these plots (Figure 8,9,10) in order to make this section flow smoothly and not feel as choppy. I would also reorganize the results about figure 10 and the reservoir location to be next to the map.
  **Reply:** *Thank you for these great suggestions. We created two new figures that combine the contents of previous figures 7-10. The new Figure 7 combines the results previously shown in Figures 7 and 10 and the new Figure 8 combines the results previously displayed in Figures 8 and 9.*

Discussion:

- I do not feel that you did a strong job of linking catchment elevation to reservoir operations, Perhaps that is due in part to the shorter results section. I think main use is a bigger takeaway as you specifically state that higher elevation reservoirs are more hydropower vs lower elevation being water supply.
  **Reply:** *We added a summary of the reservoir purposes of catchments in clusters 1 and 2 to the results section (see Figure 5 in this response to the reviewer file). The figure shows that reservoirs in the (high-elevation) catchments in Cluster 1 are primarily used for energy production, while the reservoirs in catchments belonging to Cluster 2 are operated for a variety of purposes including energy production, flood control, and recreation.*

[Figure]

*Figure 1: Reservoir purpose mix of catchments in regulation clusters 1 and 2 (see Figure 9 in the paper): Energy production, flood control, and recreational use.*

**Minor Edits:**

- Figure 2: I would add a dashed line when the reservoir came online so we have a better idea of when those changes occurred.
  **Reply:** *We added a line to the plot indicating when the reservoir was constructed.*

- Figure 4: the grey on the natural flow is hard to see. Perhaps using a dashed line or something like that would be useful?
  **Reply:** *We changed the color to a darker grey to increase visibility.*

- Figure 3: I really liked this figure!
  **Reply:** *Thank you.*

**Notes from HESS specific questions:**

1. Does the paper address relevant scientific questions within the scope of HESS?

Yes. It looks at how humans have impacted the hydrology of certain regions by deriving reservoir operations from generalized models

2. Does the paper present novel concepts, ideas, tools, or data?

I believe it does, but I think the authors could emphasize why this is so novel.
**Reply:** *We reworked the introduction to highlight the two novelties of our paper: (1) we propose a statistical three-step approach for reservoir signal reconstruction in catchments where reservoir outflow but no inflow time series are available. (2) The combination of the proposed reservoir signal reconstruction approach with functional clustering allows us to provide insights into how reservoir regulation varies spatially in the Alps and to which degree these variations are related to catchment characteristics.*

3. Are substantial conclusions reached?

I personally felt that the conclusions reached could have been more direct

**Reply:** *We separated the conclusions from the discussion section to highlight the main conclusions: 'We find that in the Central Alps there are two groups of catchments with distinct reservoir operation strategies: high-elevation catchments with pronounced seasonal water redistribution from summer to winter for hydropower production and low-elevation catchments with weak seasonal water redistribution for different reservoir purposes. The reservoir signals reconstructed using the GAM modelling approach may be used to inform hydrological model development and calibration. Furthermore, the reconstructed signals could inform the representation of reservoir operation in hydrological models. Improving such representation is crucial to advance the field of change attribution as it will allow for a better separation of climate and regulation signals, which both influence streamflow characteristics.'*

4. Does the title clearly reflect the contents of the paper?

I think there could be a more informative title (something like "Deriving reservoir operations from streamflow using GAMS), because you make the case in your introduction that your main takeaway is the ability to derive reservoir operations directly from streamflow, climate, etc

**Reply:** *Thank you for this alternative title suggestion. We changed the title to: 'Spatial variability in Alpine reservoir regulation: deriving reservoir operations from streamflow using GAMs'*

---

## Author Response (AR2)

**Dear Micha,**

Thank you very much for reassessing our manuscript. We appreciate the additional suggestions provided by the two reviewers and have adjusted the manuscript accordingly. As suggested by Reviewer 1, we provide a more rigorous evaluation of the approach for ten (instead of one) catchments in Switzerland, for which data on regional reservoir storage are available. In addition, we addressed the two minor points raised by the two reviewers. We hope that you find the revised manuscript suitable for publication in Hydrology and Earth System Sciences.

**Best regards,**

Manuela and Philippe

**Reviewer 1**

I would like to commend the authors on the quality and depth of their revision. I only have one major concern, and a minor suggestion. **Reply:** *Thank you very much for taking the time for your re-assessment.*

**Major concern:**

This regards the validation. The point about the difficulty of obtaining records is well-taken, and the strategy to make up for data deficiencies (using canton-level data from Switzerland) is really interesting. Yet, only anecdotal evidence is provided that the validation strategy could work, with data from a single reservoir within a single canton. A more robust validation strategy is needed, not least to alleviate concerns about cherry-picking of the data for validation (take one reservoir for which the validation strategy works). What is more, it is not clear how reservoir-level curves and canton-level curves should necessarily correspond: the method could be perfectly sound and there might be reservoirs in a canton with a very different storage signature than the average. My suggestion would be to compare the canton-level data with an average of reservoirs that are both in the canton, and included in the dataset. Please do that for all cantons for which you have enough data.

**Reply:** Thank you for this great suggestion of how to make our evaluation more robust. We redid the evaluation for ten catchments in Switzerland located in the cantons Valais, Grisons, and Ticino, for which regional reservoir storage curves are available. We adjusted Figure 7 (see Figure 1 in this response to the reviewer) with the updated results and modified the text as follows: 'A direct evaluation of the extracted seasonal reservoir signals is unfortunately not possible because observed inflow and outflow data are not publicly available. Therefore, we evaluated the approach using an alternative validation strategy. The Swiss Federal Office of Energy provides weekly reservoir storage estimates aggregated over a larger region (i.e. canton). We use these regional storage estimates to compute seasonal changes in regional storage. We then use the regional storage change curve derived for the regions Valais, Grisons, and Ticino to evaluate the reservoir signals extracted using the GAM for all catchments located in the three regions (Figure 7). That is, we apply the GAM approach to the catchments located in the cantons Valais, Grisons, and Ticino using temperature, precipitation, and glacier mass balance changes as covariables. We then compare the extracted reservoir regulation signals to the reservoir signals extracted from the regional storage curves. The regulation signal storage for the ten catchments using the GAM approach compare very well to the

**signals derived from observed regional reservoir storage data.'**

Figure 1: Observed reservoir storage change curves for the regions (a) Valais, (b) Grisons, and (c) Ticino derived from regional reservoir storage data provided by the Swiss Federal Office of Energy compared to the reservoir regulation signals estimated using the GAM of the catchments located in the corresponding cantons (d): (a) Rhône, Porte-du-Scex; Rhône, Sion; Rhône, Branson; Drance de Bagnes, Le Châble; and Vispa, Visp, (b) Inn, Tarasp; Inn, Martina; and Spöl, Punt dal Gall, and (c) Brenno, Loderio and Moesa, Lumino.

**Minor suggestion:**

In Table 1A, it would be great to either clarify that all records stretch to the present in the caption (and clarify that 2022 is the present year at time of writing), or if that's not the case, add a column with the start date of the record.

**Reply:** We added a new column to Table 1A, that specifies the start date of all the records.

**Reviewer 2**

My only edit up on the receipt of the revised draft is to change the background on figure 3 to be white and change the box color of the "simulated natural time series" to a less bright color as it clashes with the other box.

**Reply:** Thank you very much for your re-assessment, which we highly appreciate. We changed the background color of Figure 3 to white and adjusted the color for 'simulated natural time series' to a less bright red, which was also used in Figures 4 and 5.